# Do Media Extracellular Vesicles and Extracellular Vesicles Bound to the Extracellular Matrix Represent Distinct Types of Vesicles?

**DOI:** 10.3390/biom14010042

**Published:** 2023-12-28

**Authors:** Saida Mebarek, Rene Buchet, Slawomir Pikula, Agnieszka Strzelecka-Kiliszek, Leyre Brizuela, Giada Corti, Federica Collacchi, Genevieve Anghieri, Andrea Magrini, Pietro Ciancaglini, Jose Luis Millan, Owen Davies, Massimo Bottini

**Affiliations:** 1Institut de Chimie et Biochimie Moléculaires et Supramoléculaires, UMR CNRS 5246, Université de Lyon, Université Claude Bernard Lyon 1, 69 622 Villeurbanne Cedex, France; renebuchet@bluewin.ch (R.B.); leyre.brizuela-madrid@univ-lyon1.fr (L.B.); 2Laboratory of Biochemistry of Lipids, Nencki Institute of Experimental Biology, Polish Academy of Sciences, 3 Pasteur Street, 02-093 Warsaw, Poland; s.pikula@nencki.edu.pl (S.P.); a.strzelecka-kiliszek@nencki.edu.pl (A.S.-K.); 3Department of Experimental Medicine, University of Rome Tor Vergata, 00133 Rome, Italy; giada.corti@alumni.uniroma2.eu (G.C.); federica.collacchi@alumni.uniroma2.eu (F.C.); 4School of Sport, Exercise and Health Sciences, Loughborough University, Loughborough LE113TU, UK; g.anghileri@lboro.ac.uk (G.A.); o.g.davies@lboro.ac.uk (O.D.); 5Department of Biomedicine and Prevention, University of Rome Tor Vergata, 00133 Rome, Italy; andrea.magrini@uniroma2.it; 6Departamento de Química, Faculdade de Filosofia, Ciências e Letras de Ribeirão Preto, Universidade de São Paulo, Ribeirão Preto 14040-901, São Paulo, Brazil; pietro@ffclrp.usp.br; 7Sanford Children’s Health Research Center, Sanford Burnham Prebys, La Jolla, CA 92037, USA; millan@sbpdiscovery.org

**Keywords:** media extracellular vesicles, matrix vesicles, cell-cell communication, biomineralization, collagenase, ultracentrifugation

## Abstract

Mineralization-competent cells, including hypertrophic chondrocytes, mature osteoblasts, and osteogenic-differentiated smooth muscle cells secrete media extracellular vesicles (media vesicles) and extracellular vesicles bound to the extracellular matrix (matrix vesicles). Media vesicles are purified directly from the extracellular medium. On the other hand, matrix vesicles are purified after discarding the extracellular medium and subjecting the cells embedded in the extracellular matrix or bone or cartilage tissues to an enzymatic treatment. Several pieces of experimental evidence indicated that matrix vesicles and media vesicles isolated from the same types of mineralizing cells have distinct lipid and protein composition as well as functions. These findings support the view that matrix vesicles and media vesicles released by mineralizing cells have different functions in mineralized tissues due to their location, which is anchored to the extracellular matrix versus free-floating.

## 1. Introduction

Most cells and tissues release nanoparticles and microparticles which can be vesicular [1,2,3,4,5,6,7,8] or non-vesicular [9,10,11]. Their lipid and protein composition and functions may differ depending on the cell type and physiological or pathological conditions [12,13,14,15,16,17,18,19]. There is a consensus to classify vesicular particles (extracellular vesicles) based on their size, density, or mechanism of biogenesis [20,21]. Small-sized vesicles have a size ranging from 40 to 150 nm (sometimes referred to as exosomes), medium-sized vesicles have a size ranging from 100 to 300 nm, while large-sized vesicles can have a size up to several microns (this class includes apoptotic bodies) [21]. Extracellular vesicles may originate from the endosomal network (exosomes) or through the budding of the plasma membrane (ectosomes) [8]. Non-vesicular nanoparticles, including exomeres (smaller than 50 nm) [10] and supermeres [11], are also found in the extracellular medium. Supermeres are separated from exomeres after high-speed ultracentrifugation of supernatants [11]. Extracellular vesicles can interact with the extracellular matrix and modulate its structure and function [22,23]. Herein, we will discuss a few examples of extracellular matrix-bound vesicles (matrix vesicles), how they can be extracted, and how they are different from the extracellular vesicles unbound to the extracellular matrix (media extracellular vesicles extracted directly from the extracellular medium or, more simply, media vesicles).

## 2. Discovery of Matrix Vesicles

Lipid components in the mineralizing front of cartilage were revealed by Sudan Black B staining of growth plate cartilage [24,25,26]. Later, electron microscopy indicated the presence of 100–300 nm in diameter vesicular structures at the site of epiphyseal cartilage in mice [27,28]. Cartilage at an early stage of calcification of 1-month-old guinea pigs and at proximal tibial-distal femoral epiphyses of 3-day-old rats showed the presence of roundish bodies, which gradually become filled with crystallites [29]. The first extraction of mineralizing vesicles was carried out using bovine fetal or rabbit epiphyseal cartilages [30,31]. A collagenase digestion of the epiphyseal cartilage, followed by several differential centrifugations, was performed [30,31,32]. An enriched amount of cholesterol, phosphatidylserine, and sphingomyelin was found in mineralizing vesicles, as compared to the composition of plasma membranes [33,34,35]. The isolated vesicles had high tissue-nonspecific alkaline phosphatase (TNAP) activity [32,36,37]. The first mineralizing vesicles were isolated directly from cartilage tissues and were released after collagenase digestion. They were not extracted from either the extracellular medium or biological fluids. At that time, mineralizing vesicles were referred to as matrix vesicles or collagenase-released vesicles [38]. Since their initial discovery in the growth plate cartilage, other mineralizing vesicles have been found at the first mineral deposit site during intramembranous bone formation [39], in fracture callus [40], in developing dentin [41], in pathological calcification of valves [42], and in osteosarcoma [43].

## 3. Discovery of Media Vesicles

The discovery of media (extracellular) vesicles has been reviewed elsewhere [20,44,45,46]. Here, we briefly communicate work describing the occurrence of extracellular vesicles not bound to the extracellular matrix. Early work on the clotting factors in human plasma indicated the presence of blood corpuscles, in addition to the thromboplastic agent, sedimented at 31,000× *g* [47]. The material extracted from plasma and separated by ultracentrifugation was enriched in phospholipids and showed coagulant properties resembling those of platelet factor 3 [48]. Around 1960, several pieces of evidence suggesting the occurrence of extracellular vesicles in platelets [48] secreted by mammalian cells [49], as well as non-mammalian phagocytic cells [50,51] were collected by means of electron microscopy. Extracellular synaptic vesicles at sites of the periaxonal space within the mouse atrium were evidenced by electron microscopy [52,53]. It was also discovered that extracellular vesicles can contribute to neuronal signaling [54,55]. Addition of the A23187 cation ionophore to human red blood cells induced a discocyte to echinocyte morphological change and the release of extracellular vesicles enriched in 1,2-diacylglycerol [56]. Extracellular vesicles released from *Ochromonas danica* were evidenced by electron microscopy in the early 1970s [57]. It was not a fixation artifact since extracellular vesicles could be isolated [57]. Similarly, other microorganisms such as *Candida* [58], *Corynebacterium* [59], *Acinetobacter* [60], and *trypanosoma cruzi* [61] can release extracellular vesicles. The first indication that these particles could mediate functional biological effects was indicated by the discovery that major histocompatibility complex (MHC) class II-containing extracellular vesicles from B lymphocytes could regulate the activity of T cells [62]. Later, horizontal RNA transfer was reported between extracellular vesicles and recipient cells [63,64]. Extracellular vesicles were referred to by several names around this time by groups working in different fields and it was often unclear exactly what these particles were [20]. In this respect, it is important to emphasize that exosomes are generated via the endocytic pathway, which correspond to one subcategory of extracellular vesicles [65,66]. Another category of extracellular vesicles which shed directly from plasma membranes are called ectosomes [20]. Since the sizes of ectosomes and exosomes may overlap and sometimes no specific markers have been identified, they are collectively referred as extracellular vesicles [20].

## 4. How to Differentiate Matrix Vesicles and Media Vesicles

In this section, we describe the model showing that not all types of extracellular vesicles can be found in the extracellular medium, but a distinct population remains strongly bound to the extracellular matrix. The model can be validated by comparing the lipid and protein composition and functions of the extracellular vesicles isolated directly from the extracellular medium (media vesicles) with those of the extracellular vesicles isolated after discarding the extracellular medium and subjecting the cells and/or tissues to an enzymic digestion (matrix vesicles) (Figure 1).

There is no perfect extraction method to isolate extracellular vesicles, and therefore the isolation step should be accompanied by a biochemical analysis to fully characterize the extracellular vesicles and their functions [21,67]. Several pieces of evidence from studying mineralizing cells support the model that media vesicles are distinct from matrix vesicles [38,68]. In the following sections, we will compare the properties of media vesicles and matrix vesicles released from chondrocytes, osteoblasts, and smooth muscle cells.

Media vesicles have been isolated by using a variety of published protocols [69] (Figure 1A). Ultracentrifugation, polyethylene glycol precipitation, total exosome isolation reagent, and an aqueous two-phase system with and without repeat washes or size exclusion chromatography have been assessed for their ability to extract extracellular vesicles [70]. Among these methods, size exclusion chromatography and ultracentrifugation were favored for overall efficiency [70]. Although there are variations in sample purity between isolation methods, scalability, and yield, ultracentrifugation represents the most common method but remains not specific. Further specificity could be gained after ultracentrifugation by using affinity chromatography or a combination of size exclusion chromatography with ultrafiltration to maximize both yield and purity [71]. One of the hallmarks of the mineralization process is the presence of tissue non-specific phosphatase (TNAP) activity of the cells and of extracellular vesicles [38]. The expression of TNAP in media vesicles and in matrix vesicles could be compared to further substantiate their differences. In this paragraph, we briefly describe the ultracentrifugation method since it is commonly applied for the recovery of both media and matrix vesicles. Briefly, extracellular medium is harvested and centrifuged at 1000× *g* for 30 min at 4 °C to remove cells and bigger debris. A second centrifugation, which is optional, can be performed at 20,000× *g* for 30 min to isolate large-sized media vesicles along with smaller debris (smaller debris can be removed by a successive step based, for instance, on chromatography). The final centrifugation step is performed at 100,000× *g* for a time ranging from 30 min to 2 h. The pellet obtained at this stage contain small- and medium-sized media vesicles. The pellet is resuspended preferably in ice-cold synthetic cartilage lymph, which is a buffer matching the electrolyte composition of the extracellular milieu in cartilage and bone tissues. This is not the case for most media vesicle studies. However, buffer composition has been optimized to obtain the most stable matrix vesicles [72]. To compare media vesicles and matrix vesicles, the same buffer should be used. In this respect, most of the findings concerning the comparisons of properties of media vesicles and matrix vesicles are from mineral-competent cells. The synthetic cartilage lymph contains 1.42 mM Na_2_HPO_4_, 1.83 mM NaHCO_2_, 12.7 mM KCl, 0.57 mM MgCl_2_, 100 mM NaCl, 0.57 mM Na_2_SO_4_, 5.55 mM glucose, 63.5 mM sucrose, and 16.5 mM 2-{2-hydroxy-1,1-bis (hydroxymethyl) ethyl} amino)-propanesulfonic acid (pH 7.4) [72,73].
Figure 1(**A**) Isolation of media vesicles. Cells and the extracellular matrix are discarded. The extracellular medium is subjected to two–three centrifugation steps to obtain either the large-sized vesicles or small- and medium-sized vesicles. Extracellular vesicles anchored to the collagenous matrix (i.e., matrix vesicles) remain mostly attached to the extracellular matrix and only a small population can be extracted. (**B**) Isolation of matrix vesicles. The extracellular medium is discarded. Cells and the extracellular matrix are washed with synthetic cartilage lymph [72,73]. An enzymatic digestion, for instance, with collagenase, is performed to degrade the collagen fibers and release matrix vesicles. After several centrifugation steps, matrix vesicles are obtained. * = optional step. Abbreviations: SUR = supernatant; PEL = pellet.
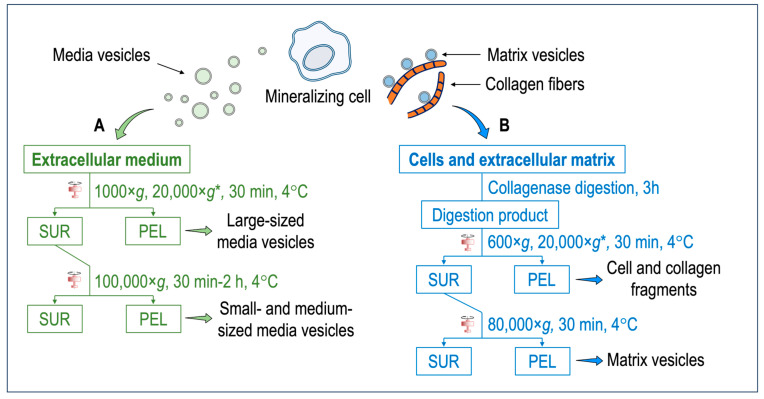



Most matrix vesicles with high TNAP activity are released from fully differentiated and mineral competent cells, while undifferentiated cells release extracellular vesicles with little TNAP activity [68]. Matrix vesicles are released from hypertrophic chondrocytes [28,28,74,75,76,77]. Hypertrophic chondrocyte differentiation is associated with high TNAP activity, and the synthesis and secretion of type X collagen followed by type II collagen by proliferating and pre-hypertrophic chondrocytes [28,74,75,76]. Expression of type I collagen by hypertrophic chondrocytes might be associated with differentiation into osteoblast-like cells [77,78,79]. Matrix vesicles correspond to mineralizing cartilage, where the growing cartilage is replaced by bone, while articular cartilage matrix vesicles originate from normal cartilage that does not undergo matrix mineralization except in pathologic conditions such as osteoarthritis [80]. Articular matrix vesicles from normal cartilage show low TNAP activity, however during osteoarthritis chondrocytes can become hypertrophic and fully mineralized. They release articular matrix vesicles which can induce pathologic calcium crystal deposition in articular cartilage matrix [81]. The release of matrix vesicles by osteoblasts is stimulated in osteogenic medium containing ascorbic acid and beta-glycerophosphate [82]. TNAP activity of Saos-2 cells increased with the duration of the treatment with osteogenic factors which coincided with the amount of released matrix vesicles [83]. Matrix vesicles have been isolated by the collagenase-digestion method [38,84,85] (Figure 1B). Briefly, tissues (for instance, growth plates and epiphyseal cartilage from 17-day-old chicken embryos) are cut into 1–3 mm thick slices and washed with synthetic cartilage lymph. In the case of cells (for instance, osteoblast and chondrocyte cultures), the extracellular medium is discarded, and the cells embedded in the collagenous matrix are washed with synthetic cartilage lymph. The osseous and cartilage tissues or cells are vortexed in synthetic cartilage lymph containing 1 mM CaCl_2_ and 100–200 U of type-I collagenase from *Clostritidium histolyticum* at 37 °C for 180 min. Since the quality of the commercial collagenase is variable, the amount of the collagenase to be adjusted can be determined by measuring TNAP activity of the matrix vesicles [84]. Indeed, scientists have taken advantage of this to define matrix vesicle purity as having TNAP activity that is a minimum of two-fold greater than that of the plasma membrane fraction [86]. From this stage, either a three-step centrifugation [87] or a two-step centrifugation [88] is usually performed. The collagenase-digested sample is filtered through a nylon filter and centrifuged at 600× *g* for 30 min at 4 °C to remove cells and fragments of the extracellular matrix. The pellet is discarded, and the supernatant is subjected to an optional centrifugation step at 20,000× *g* for 30 min at 4 °C to remove membrane debris. Then, the supernatant is ultracentrifuged at 80,000× *g* for 60 min at 4 °C. The final supernatant is removed, while the pellet, containing matrix vesicles, is washed three times with synthetic cartilage lymph to remove collagenase and calcium. Matrix vesicles shall not be frozen—however they can be stored at 4 °C up to five days [84] to preserve their original enzymatic activity, including those of TNAP, P_i_ and Ca transporters, and other membrane proteins. The freezing thaw process can induce membranous defects which can leak ions and/or other soluble particles. Cryoprotectant, as sucrose or trehalose, can be added [89] to preserve the integrity of extracellular vesicles [89]. The collagenase-digestion method provides a relatively high amount of matrix vesicles displaying a TNAP specific activity of around 15 to 30 μmole min^−1^ mg^−1^ [84]. Alternatively, trypsin [90], trypsin/collagenase [91], liberase/blendzyme-1 [92], hyaluronidase [80,85], and hyaluronidase/collagenase [91] can be used to release matrix vesicles from the extracellular matrix (Table 1). Collagenase digestion is among the most used digestion method to isolate matrix vesicles which can yield a high ratio of alkaline phosphatase activity in matrix vesicles compared to that in media vesicles or plasma membrane (Table 1). The collagenase and hyaluronidase digestion, which removed the surface collagens on the matrix vesicles, induced a loss of calcium uptake [91] (Table 1). In contrast collagenase and trypsin digestion, which maintained a part of surface collagens on matrix vesicles, preserved the calcium uptake [91] (Table 1). The process of trypsin digestion resulted in the release of matrix vesicles from the extracellular matrix, exhibiting an alkaline phosphatase activity six times greater than that of the plasma membrane. [93] (Table 1). Liberase and blendzyme1, which is gentle digestion, was the less efficient method to release matrix vesicles than the other digestion methods, as indicated by the lower ratio of alkaline phosphatase in matrix vesicles as compared to that in media vesicles [92] (Table 1).

Although collagenase digestion is often used to release the cells from the extracellular matrix, enzymatic digestion may alter/damage surface protein expression of the matrix vesicles. For instance, the main collagen identified in matrix vesicles from growth plate cartilage is collagen type VI, which is consistent with the resistance of this type of collagen to the collagenase digestion [94]. Only three peptides corresponding to cartilage specific collagen type X are identified from proteomic analysis [94]. Generally, the quality of matrix vesicles is assessed by TNAP activity, an ability to form apatites inside MVs, turbidity measurements, and morphological assessment by electron microscopy [38,84]. As a result of matrix vesicles not being lysed during isolation, their native conformation remains intact. Thus, they remain right side out, with the ectoenzyme TNAP facing outward [85]. Each digestion method displays a different efficiency to release matrix vesicles and/or maintain their physical and biochemical properties [91].

One essential function of matrix vesicles is to initiate apatite formation and to deposit it onto collagen fibers [38,95,96]. Several pieces of evidence support that matrix vesicles are strongly bound to the extracellular matrix. Northern blot and immunohistochemical analyses on matrix vesicles from chondrocytes revealed an increase in annexin A5 and type I collagen [76]. Annexin A5 is highly enriched in matrix vesicles and was found to bind to native type I, II, and X collagens [97,98,99,100]. Neutral metalloproteases 2, 9, and 13 [101,102] were found in matrix vesicles, which would suggest that the initiation of matrix vesicle-induced mineralization is coupled with the degradation of the inhibitory proteoglycan matrix [38]. Matrix vesicles are anchored within the extracellular matrix via integrin binding to type II collagen [103].

In the next sections, we will ask the question whether there are differences on composition and functions of media vesicles and matrix vesicles released by mineralizing cells. An advantage to focus essentially on the extracellular vesicles released from mineralizing cells is the possibility to discriminate the two classes of vesicles by their ability to accumulate apatite in their lumen.

## 5. Matrix Vesicles and Media Vesicles from Growth Plate Cartilage and Hypertrophic Chondrocytes

Chick growth plates are often used to obtain a large amount of matrix vesicles [38]. Collagenase-released vesicles are distinct from the vesicles isolated without collagenase treatment [104]. Growth plate chondrocytes do produce vesicles typical of exosomes that they release into the culture media [105]. However, they differ significantly from matrix vesicles [85] as they are not enriched in alkaline phosphatase activity, and possess different enzymes than are found in matrix vesicles [85]. Matrix vesicles, unlike exosomes, are anchored to the extracellular matrix [85]. Without enzymatic digestion, at least four types of media vesicles can be released from the growth plate cartilage, as evidenced by sucrose gradient ultracentrifugation (Figure 2A) [94]. Conversely, collagenase digestion of the growth plate cartilage yields only two lighter fractions which were identified as matrix vesicles. Matrix vesicles have the lowest density (ρ = 1.12–1.14 g.cm^−3^), the highest lipid to protein ratio (around 2–3 mg/mg), and the highest TNAP specific activity (Figure 2B) [94]. Media vesicles, extracted without collagenase treatment, have detectable TNAP activity but they do not induce mineral formation well. Matrix vesicles and media vesicles have distinct protein profiles (Figure 2C) [94].

Earlier findings indicated that the (media) vesicles in the fractions obtained without collagenase treatment and isolated from sucrose gradient were distinct from the collagenase-released (matrix) vesicles [104]. The osmotic pressure induced by sucrose gradient could affect the mineral property of matrix vesicles due to the loss of ability to accumulate Ca and P in the lumen [38,104]. Percoll gradient isolation enabled researchers to obtain a fraction with two major populations of matrix vesicles with one having mineralizing functions and the other not [106], however, this method is rarely used. Therefore, the experimental evidence that matrix vesicles can accumulate Ca and P in the lumen was obtained from vesicles extracted after collagenase digestion and centrifugation but without using sucrose gradient (Figure 2B). Transmission electron microscopy coupled with energy dispersive X-ray spectroscopy of matrix vesicles indicated that the Ca/P ratio in neat matrix vesicles (that is, in the absence of extracellular Ca^2+^) was lower than that in the vesicles in the presence of extracellular Ca^2+^ [107], which confirmed the calcium influx inside the vesicles (Figure 3A–C). Infra-red analysis of mineral deposits inside matrix vesicles indicated the presence of apatite (Figure 3D).

Matrix vesicles have high levels of cardiolipin and sphingomyelin compared to the plasma membrane [34], while phosphatidylserine and phosphatidylinositol are enriched in the inner leaflet of the phospholipid bilayer [32,108].

As an alternative to growth plate cartilage, chondrocyte cultures [74,86,109,110,111,112,113,114,115,116,117,118,119,120,121,122,123] can be used as a starting material to compare the properties of media vesicles and matrix vesicles. Chondrocytes isolated from the proliferating and hypertrophic regions of the growth plate differ significantly in their ability to release TNAP-rich matrix vesicles [105]. Functional matrix vesicles with high TNAP specific activity are usually extracted from hypertrophic chondrocytes [38]. To stimulate cell differentiation, 50 μg·mL^−1^ ascorbic acid and 10 mmol.L^−1^ β-glycerophosphate are added to the primary chondrocytes isolated from the articular cartilage of 4–6 days old mice [124]. β-glycerophosphate, often used to stimulate mineralization of chondrocytes in culture, does not correspond to physiological conditions. As an alternative, 2, 4, 7 or 10 mM inorganic phosphate can be added to stimulate mineralization in chondrocytes [125]. From this point, hypertrophic chondrocytes are washed with synthetic cartilage lymph and treated with collagenase followed by several differential centrifugations to obtain matrix vesicles (Figure 4A, left side). The discarded extracellular medium is subjected to several centrifugation steps to obtain media vesicles, including apoptotic bodies and exosomes (Figure 4A, right side). It is worth noting that matrix vesicles from primary hypertrophic chondrocytes lack ALIX and CD9 (Figure 4B), however CD9 was detected in matrix vesicles from growth plate cartilage by proteomic analysis [94]. Matrix vesicles are distinct from small-and medium-sized media vesicles enriched in ALIX and CD9, which were isolated from the same hypertrophic chondrocytes (Figure 4B, top panel). Large-sized vesicles (apoptotic bodies) contain calnexin (Figure 4B, top panel). In general, media vesicles are not homogeneous since they may contain variable exosomal and non-exosomal subpopulations [126]. We can’t exclude that matrix vesicles contain other subpopulations [85]. TNAP activity is high in matrix vesicles in contrast to exosomes or other vesicles (Figure 4B, bottom panel). It was earlier reported that apoptotic bodies and matrix vesicles have distinct mineral forming property [127]. In contrast to matrix vesicles, media vesicles are less able to induce mineral formation, regardless of size, as indicated by the turbidity (Figure 4C). There is the possibility that a small fraction of matrix vesicles and/or other types of mineralizing vesicles could be present in media vesicles despite the low TNAP activity due to the difficulty to obtain a pure fraction of one type of extracellular vesicles. It was earlier reported that microvilli are the precursors of matrix vesicles, and that retraction of the supporting microfilament network is essential for the release of these structures [115] while exosomes originate from an endosomal pathway [128].

## 6. Matrix Vesicles and Media Vesicles from Differentiated Osteoblasts

Human osteosarcoma Saos-2 cells are often the cellular source to extract matrix vesicles [83,129,130,131,132,133] since they can produce matrix vesicles and mineralize the extracellular matrix [134]. Other osteoblast-like cell lines include rat osteosarcoma ROS 17/2.8 cells [82,110,135,136,137], C57BL/6 mouse calvaria MC3T3 (cells or sub-line MC3T3-E1) [82,92,110,135,136], human osteosarcoma U2-OS cells (Jiang et al. 2013), human osteosarcoma MG-63 cells [82,110,135,136,137], and human fetal osteoblast hFOB [131]. Matrix vesicles are extracted from mineralizing osteoblasts fully differentiated after two–three weeks of incubation in an osteogenic medium containing ascorbic acid (50–100 μg/mL) and β-glycerophosphate (7.5–10 mM) [82]. Indeed, extracellular vesicles derived from non-mineralizing osteoblasts were not found to enhance mineralization in human bone marrow-derived mesenchymal stem cells in contrast to those extracted from mineralizing osteoblasts stimulated by exogeneous phosphate [138]. Not only osteogenic medium may affect the mineral properties of cells. Inflammatory cytokines can induce mineralization by influencing the enzymes regulating the pyrophosphate (inhibitor of mineralization) to phosphate ratio [81]. Extracellular vesicles produced by IL-1β treated mesenchymal stem cells have altered ratio of ectonucleotide pyrophosphatase/phosphodiesterase 1 to alkaline phosphatase and less pyrophosphate in the vesicle fraction [139]. Calcium nodules characteristic for osteoblastic mineralization can be detected by means of Alizarin Red-S staining after 6–12 days of incubation in an osteogenic medium (Figure 5A,B). TNAP specific activity is higher in Saos-2 cells stimulated with 50 µg/mL ascorbic acid and 7.5 mM β-glycerophosphate than in unstimulated cells (Figure 5C).

The accumulation of minerals inside the matrix vesicles embedded in bone tissues has been observed using electron microscopy [140,141,142]. Transmission electron microscopy coupled with X-ray microanalysis spectroscopy (TEM-EDS) of matrix vesicles released from Saos-2 cells indicated that Ca and P ions or elements co-localized inside matrix vesicles ([131], Figure 6).

Several pieces of evidence indicate that matrix vesicles secreted by osteoblast-like Saos-2 cells bud off from the plasma membrane, especially from microvilli [128]. Cytochalasin D, which inhibits actin polymerization, stimulates cell apoptosis and facilitates the release of matrix vesicles [83]. In contrast, phalloidin, which stabilizes actin filaments, inhibits matrix vesicle secretion, indicating that matrix vesicle secretion is induced by actin depolymerization [83]. 93% of the proteins (262 proteins over 282) found in matrix vesicles from Saos-2 cells are present in microvilli-like membranes from Saos-2 cells [130]. Of particular interest is that β-actin is always found in matrix vesicles from growth plate cartilage [94], MC3T3-E1 cells [92], and Saos-2 cells [129], while β-actin is absent in media vesicles from MC3T3-E1 cells [92] and in primary calvaria osteoblasts [144]. Matrix vesicles and microvilli are enriched in cholesterol, phosphatidylserine, sphingomyelin, while they are depleted in phosphatidylcholine and triacylglycerols in contrast to basolateral membranes [129]. Leucine aminopeptidase (microvilli marker), annexin A2, annexin A6, Na^+^/K^+^ ATPase, and TNAP are highly enriched in both matrix vesicles and microvilli as compared to basolateral membranes [129]. Taken together, these findings support that matrix vesicles bud from the plasma membrane. Initially, CD9 was considered an exosomal marker but later it was proposed that CD9 could be linked to budding vesicles [145]. CD9 was found by proteomic analysis in matrix vesicles from MG-63 osteoblasts [93] and in matrix vesicles from the growth plate cartilage [88], while it was undetected in matrix vesicles from MC3T3-E1 cells [92] and Saos-2 cells [129]. Exosomal marker CD81 was also present in matrix vesicles from MG-63 osteoblasts [93] and Saos-2 cells [129] but it remained undetected in matrix vesicles from MC3T3-E1 [92] and the growth plate cartilage [94]. The apparent contradiction lies from the fact that while various types of extracellular vesicles contain several common markers, including tetraspanins, their relative proportions vary in the different vesicle types [146]. Alternatively, since the isolation methods are not optimal, other types of extracellular vesicles may coexist. In this respect, the quantification of the occurrence of the different markers could contribute to determine the populations of different subtypes of extracellular vesicles. Alternatively, the development of more selective methods to fully characterize the subpopulations of media vesicles and of matrix vesicles are needed. Nevertheless, matrix vesicles appear to be distinct from media vesicles due to their distinct TNAP specific activity and different lipid and enzyme content [93]. Matrix vesicles and media vesicles serve a different role in bones due to their location within mature bone, as they are anchored to the extracellular matrix versus free-floating [93]. Media vesicles can communicate with distant cells, while matrix vesicles remain close to the parent cells [93]. Consistent with this view, local administration of matrix vesicles from mouse bone marrow-derived stromal cell line ST2 embedded in gelatin hydrogels restored the femoral bone defect in mice [147]. In contrast, media vesicles secreted from either primary osteoblast or MC3T3-E1 cells can inhibit bone formation and enhance bone resorption through osteoblast to osteoclast communication [148]. Alternatively, media vesicles may contain a small population of matrix vesicles and/or extracellular vesicles with mineralization properties.

## 7. Matrix Vesicles and Media Vesicles from Smooth Muscle Cells

Vascular calcification occurs when vascular smooth muscle cells or circulating cells differentiate to an osteogenic-like phenotype, synthesize an extracellular matrix, and form apatite [149]. Histologic, ultrastructural, and cytochemical techniques indicated that extracellular vesicles are involved in arterial medial calcification [150] but they are distinct from their bone counterparts [151]. Extracellular vesicles released by vascular smooth muscle cells are at the sites of medial vascular calcification [152,153,154,155,156,157,158] and can be involved in atherosclerosis-related vascular calcification [151,159]. Extracellular vesicles can communicate with cells and organs to regulate vascular calcification and may serve as therapeutic methods in vascular calcification [160]. In this respect, it is essential to make a clear distinction between matrix vesicles and media vesicles. Matrix vesicles extracted after a collagenase digestion [146,161,162,163,164] and media vesicles isolated without collagenase digestion, both from vascular smooth muscle cells cultured in an osteogenic medium [146,153,165,166,167,168,169,170], have distinct characteristics both at molecular and functional levels. Proteomic profiles of matrix vesicles and media vesicles from MOVAS cell line are distinct [171]. Matrix vesicles from vascular smooth muscle cells are enriched in endosomal CD63 as compared to media vesicles, which are enriched in CD81 and CD9 [146]. Matrix vesicles induced calcification of recipient vascular muscle cells, while media vesicles appear to be less efficient [146] probably due to the enriched amount of fetuin (which inhibits calcification) in media vesicles. Media vesicles from vascular smooth muscle cells are of exosomal origin [15]. This is supported by the presence of phosphatidylserine on the outside layer leaflet of media vesicles [165]. It was proposed that phosphatidylserine exposure on the external surface of extracellular vesicles together with annexin A6 could drive the mineralization process [165], although direct experimental evidence of apatite formation induced by media vesicles released by vascular smooth muscle cells is lacking. TNAP-enriched matrix vesicles released from vascular smooth muscle cells subjected to collagenase contained more minerals with higher Ca/P ratio than the less TNAP-enriched vesicles isolated without collagenase treatment. These findings suggest a role for collagen in promoting calcification induced by TNAP in atherosclerotic plaques [172]. To explain why osteoporosis can contribute to vascular calcification—a calcification paradox—it was reported that extracellular vesicles from aged bone matrix during bone resorption can favour the adipogenesis of bone marrow mesenchymal stem/stromal cells and increase vascular calcification [173]. So far it is still unclear if such extracellular vesicles could originate from osteoclasts within the aged bone matrix. This opens the possibility that osteoclasts can release matrix-vesicles and media vesicles with distinct properties than those released from osteoblasts and from smooth muscle cells.

## 8. Concluding Remarks

Mineralization-competent cells (hypertrophic chondrocytes, mature osteoblasts, and osteogenic-differentiated vascular smooth cells) secrete matrix vesicles and media vesicles which have distinct biochemical properties (distinct protein and lipid compositions) and biological functions (ability to accumulate calcium phosphate complex in the lumen and induce calcification versus cell-cell communication). One general problem is that there is no perfect method to extract media vesicles and matrix vesicles. Indeed, several populations of extracellular vesicles may coexist in both media vesicles and matrix vesicles. So far it appears that the relative population of TNAP-enriched extracellular vesicles is distinct in media vesicles and in matrix vesicles. Matrix vesicles and media vesicles have a different role in the bone due to their location—anchored to the extracellular matrix versus free-floating [93]. Media vesicles can communicate with distant cells, while matrix vesicles remain close to their parent cells [93]. These findings support the view that matrix vesicles released by mineralization-competent cells are a specific class of vesicles which modulate the properties of the extracellular matrix, while media vesicles are more prone to participate in cell-to-cell communication to modulate cell functions. This opens the possibility that other types of cells, not only mineral-competent cells, could secrete both matrix vesicles and media vesicles, and the question of what their respective functions might be in those tissues. Extracellular vesicles may represent good candidates for tissue engineering and regenerative medicines [46,174]. In this respect matrix vesicles and media vesicles with their distinct properties derived from osteoblasts/mesenchymal stromal cells could have a considerable utility to mineralized tissue engineering. The conditions to preserve matrix vesicles and media vesicles need to be further optimized for possible therapeutical applications.

## Figures and Tables

**Figure 2 biomolecules-14-00042-f002:**
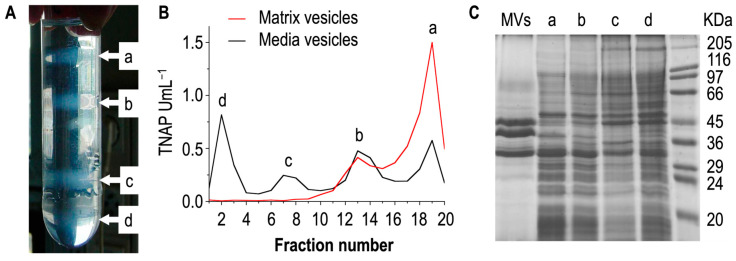
TNAP activity and protein profiles of media vesicles and matrix vesicles. (**A**,**B**) Extraction of media vesicles from growth plate cartilage of chicken embryos without collagenase digestion by a differential centrifugation in a sucrose gradient (**A**). Four fractions, labelled a (ρ = 1.12–1.14 g cm^−3^), b (ρ = 1.20–1.21 g cm^−3^), c (ρ = 1.30–1.32 g cm^−3^), and d were evidenced in media vesicles (black trace), while two fractions were extracted from growth plate cartilage of chicken embryos with collagenase digestion labelled a (ρ = 1.12–1.14 gcm^−3^) and b (ρ = 1.20–1.21 gcm^−3^) (red trace). Fraction labelled with “a” corresponds to functional matrix vesicles since they display high TNAP activity and induce mineralization. (**C**) The protein profile of matrix vesicles is distinct from each fraction of media vesicles as indicated by gel electrophoresis. Adapted from [94].

**Figure 3 biomolecules-14-00042-f003:**
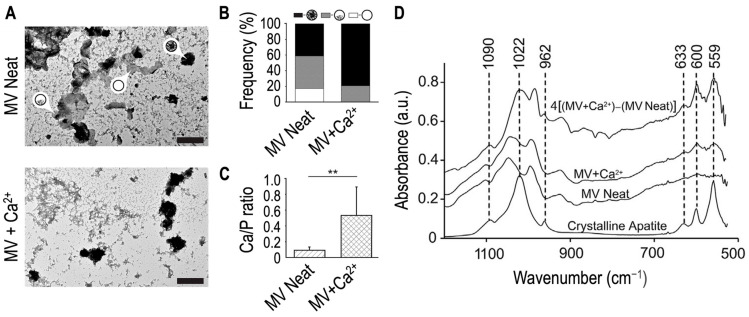
Electron microscopy and infrared analyses of matrix vesicles. Matrix vesicles were incubated in synthetic cartilage lymph devoid of Ca^2+^ (MV Neat) or supplemented with 2 mM Ca^2+^ (MV+Ca^2+^) for 24 h, then dried and analyzed by means of transmission electron microscopy coupled with X-ray. (**A**) TEM images (scale bars are 500 nm) of matrix vesicles devoid of (arrow empty circle), partially filled with (arrow partly filled circle), and fully filled with mineral deposits (arrow full filed circle). (**B**) Frequency of matrix vesicles devoid of (white area), partially filled with (grey area), and fully filled with (black area) mineral deposits. (**C**) Ca/P ratio of mineral deposits found in matrix vesicles incubated in synthetic cartilage lymph devoid of Ca^2+^ (MV Neat, hatched area) or supplemented with 2 mM Ca^2+^ (MV+Ca^2+^, cross hatched area) as measured by transmission electron microscopy coupled with energy dispersive X-ray spectroscopy. Statistical analysis was performed by Student’s t-test. ** *p* < 0.1(**D**) The top trace is the infrared spectrum for MV+Ca^2+^ after background subtraction of the MV Neat sample (middle trace) and subsequent amplification by a factor of four to better resolve the peaks. The bottom trace is the infrared spectrum of crystalline apatite as a control. Adapted from [107].

**Figure 4 biomolecules-14-00042-f004:**
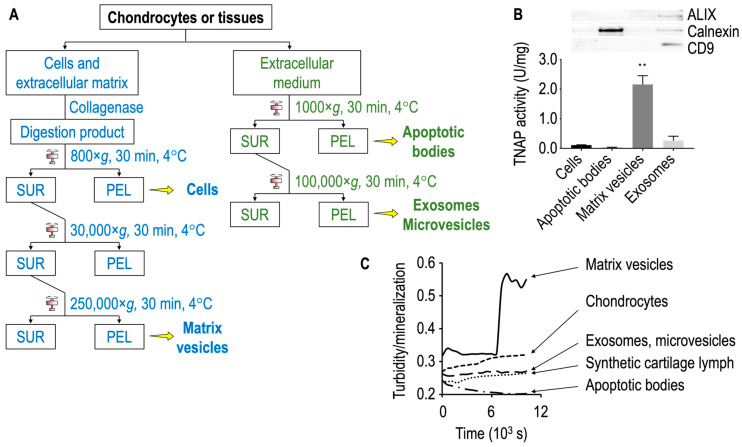
Matrix vesicles and media vesicles from primary hypertrophic chondrocytes. (**A**) Scheme of purification of matrix vesicles and media vesicles. (**B**) Top panel: Western blot of calnexin (marker of apoptotic bodies), Alix (marker of the syndecan-syntenin-Alix pathway associated with ESCRT-III), and CD9 (sometimes referred as endosomal marker). Matrix vesicles are distinct from media vesicles enriched with Alix and CD9. Bottom panel: TNAP specific activity of cells, apoptotic bodies, matrix vesicles and extracellular vesicles (mostly exosomes) indicating that MVs had the highest specific TNAP activity. Statistical analysis of TNAP specific activity of apoptotic bodies, matrix vesicles and extracellular vesicles vs cells was performed by Student’s t-test. ** *p* < 0.1. (**C**) Turbidity at 340 nm induced by the addition of 2 mM Ca^2+^ and 3.41 mM P_i_ in synthetic cartilage lymph containing either matrix vesicles, chondrocytes, or small- and medium-sized extracellular vesicles (including exosomes), or large-sized vesicles (apoptotic bodies). Only matrix vesicles induced a significant turbidity, suggesting mineral formation. (Adapted from [87]). Abbreviations: SUR = supernatant; PEL = pellet.

**Figure 5 biomolecules-14-00042-f005:**
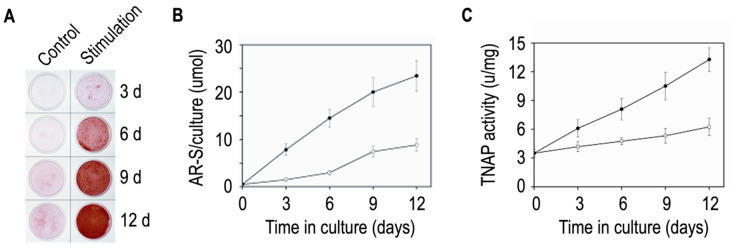
Mineralization by Saos-2 cells. (**A**) Saos-2 cells were incubated for 12 days in the absence (control) or presence of 50 µg/mL ascorbic acid and 7.5 mM β-glycerophosphate (stimulation), stained with Alizarin Red-S to detect calcium nodules and visualized under fluorescence microscope using transmitted light and phase contrast filter. (**B**) Alizarin Red-S was solubilized in control (grey trace) and stimulated (black trace) cell cultures by cetylpyridinium chloride and quantified at 562 nm (Results are mean ± SD, n = 3). (**C**) TNAP activity was calculated in control (grey trace) and stimulated (black trace) cell cultures. Adapted from [83].

**Figure 6 biomolecules-14-00042-f006:**
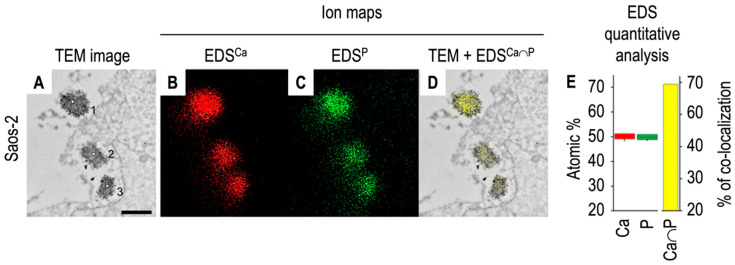
Transmission electron microscopy coupled with X-ray microanalysis of mineralizing Saos-2 cells. Saos-2 cells were stimulated for mineralization by treatment with 50 μg·mL^−1^ ascorbic acid and 7.5 mM beta-glycerophosphate for 7 days. The cells with matrix vesicles were observed under high performance TEM ((**A**), bar 500 nm) and elemental maps ((**B**), Calcium, Ca, red; (**C**), Phosphorus, P, green) were performed using EDS. Presence of calcium and phosphorus was evidenced in matrix vesicles containing dense materials. Co-localization of both elements ((**D**), yellow) and quantitative determinations ((**E**), Calcium, Ca, red, Phosphorus, P, green, Calcium to Phosphorus, yellow), provided evidence of calcium and phosphorus deposition inside vesicles, suggesting apatite deposition in matrix vesicles’ lumen. Adapted from [143].

**Table 1 biomolecules-14-00042-t001:** Properties, ratio of TNAP activity of matrix vesicles (MVs): Media extracellular vesicles (media EVs), mineralization assay, and determination of apatite in the lumen of matrix vesicles subjected to different types of enzymatic digestion. * refers to the ratio of TNAP activity of matrix vesicles:basolateral membranes and ** refers to the ratio of TNAP activity of matrix vesicles: plasma membranes. IR = infrared; ND = not determined.

Digestion	Properties	Samples	Ratio TNAP	Mineralization	Apatite in	References
Process			Activity	Assay	Lumen	
			MVs: Media EVs			
		Growth plate	From 4 to 6	YES	YES (IR)	[88]
		cartilage chicken				
Collagenase	Widely used	Primary	From 8 to 12	YES	ND	[87]
		Chondrocytes				
		Saos2 cells	Around 16 *	YES	YES (IR)	[83]
Collagenase and	MVs without	Growth plate	ND	Calcium uptake	Calcium uptake	[91]
hyaluronidase	surface collagens	cartilage chicken		was impaired	was impaired	
Collagenase and	MVs with surface	Growth plate	ND	Calcium uptake	Calcium uptake	[91]
trypsin	attached collagens	cartilage chicken		was optimum	was optimum	
Hyaluronidase	Used for	Non mineralizing	ND	ND	ND	[80]
	articular	articular				
	chondrocytes	chondrocytes				
Trypsin		MG-63 cells	Around 6 **	ND	ND	[93]
Liberase and	Gentle digestion	MC3T3-E1	From 0.7 to 0.8	ND	ND	[92]
blendzyme-1

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
