# Peer review of "Do Media Extracellular Vesicles and Extracellular Vesicles Bound to the Extracellular Matrix Represent Distinct Types of Vesicles?"

_biomolecules, 2023, doi:10.3390/biom14010042_

Round 1

Reviewer 1 Report

Comments and Suggestions for Authors

In this article, Mebarek et al reviewed and compared matrix vesicles and media vesicles from the same types of mineralized cells. They discussed a few exemples of matrix vesicles, how they can be extracted and how they are different from unbound media vesicles. They compared the properties of media vesicles and matrix vesicles released from chondrocytes, osteoblasts and smooth muscle cells. Matrix and media vesicles have distinct biochemical properties (distinct protein and lipid compositions) and biological functions (ability to accumulate calcium phosphate complex in the lumen and induce calcification via cell-cell communication). They conclude that matrix vesicles released by mineralization-competent cells are a specific class of vesicles which modulate the properties of the extracellular matrix.

This is a very extensive review and discussion of the literature concerning matrix and media vesicles derived from mineralized cells. The authors identified 169 articles of interest of which 10 are published by the authors of this review.

Overall the manuscript is well organized and well written, and I think it will be helpful for readers of Biomolecules. The authors can consider following suggestions to improve the review.

Matrix vesicles shall not be frozen to preserve their original enzymatic activity (TNAP) but can be strored at 4°C up to five days. It has been reported that lyophilization with trehalose is an effective method for the storage of vesicles for various applications. Do you know the impact of lyophilisation on matrix vesicle enzymatic activity?

Each digestion method displays a different efficiency to release matrix vesicles and/or maintain their physical and biochemical properties. The authors should add a table describing these different methods with their advantages and disadvantages.

Figure 2A/B and Figure 6 have already been published previously by the authors (Bottini et al, Biochemica Biophysica Acta 2018). 

The content of extracellular vesicles appears to be dependent on the environment (inflammation, hypoxia). Do you know if such conditions can also influence matrix vesicles and modify their enzymatic activity? The authors should comment this point.

Vascular calcification occurs when vascular smooth muscle cells or circulating cells differentiate to an osteogenic-like phenotype, synthesize an extracellular matrix and form apatite. However, vascular calcification often occurs with osteoporosis, a contradictory association called “calcification paradox”. In the study of Wang et al (Nat Com 2022), vesicles derived from aged bone matrix (AB-EVs) during bone resorption favor BMSC adipogenesis rather than osteogenesis and augment calcification of vascular smooth muscle cells. The authors should add this reference and comment matrix vesicle modifications with aging.

Extracellular vesicles, due to their favorable properties (biocompatibility, stability, low toxicity and exchange of molecular cargo) represent good candidates for tissue engineering and regenerative medicine. Matrix vesicles derived from mineralizing osteoblasts/mesenchymal stromal cells could be have considerable utility as an acellular approach to mineralized tissue engineering. The authors should comment this potential perspective.

Author Response

We would like to thank the reviewers for their fruitful comments which helped to improve greatly the manuscript. We answered point by point to each of the reviewer’s comments. The manuscript was corrected as indicated by the red-colored modifications in the manuscript.

Reviewer 1:

In this article, Mebarek et al reviewed and compared matrix vesicles and media vesicles from the same types of mineralized cells. They discussed a few examples of matrix vesicles, how they can be extracted and how they are different from unbound media vesicles. They compared the properties of media vesicles and matrix vesicles released from chondrocytes, osteoblasts and smooth muscle cells. Matrix and media vesicles have distinct biochemical properties (distinct protein and lipid compositions) and biological functions (ability to accumulate calcium phosphate complex in the lumen and induce calcification via cell-cell communication). They conclude that matrix vesicles released by mineralization-competent cells are a specific class of vesicles which modulate the properties of the extracellular matrix.

This is a very extensive review and discussion of the literature concerning matrix and media vesicles derived from mineralized cells. The authors identified 169 articles of interest of which 10 are published by the authors of this review.

Overall the manuscript is well organized and well written, and I think it will be helpful for readers of Biomolecules. The authors can consider following suggestions to improve the review.

Author’s answer: Thank you for the comments.

  • Matrix vesicles shall not be frozen to preserve their original enzymatic activity (TNAP) but can be stored at 4°C up to five days. It has been reported that lyophilization with trehalose is an effective method for the storage of vesicles for various applications. Do you know the impact of lyophilization on matrix vesicle enzymatic activity?

Author’s answer: From the early 1960, several papers reported that membrane preparation lost their enzymatic activity during the lyophilization or freeze process. Therefore, most of the findings are obtained from freshly prepared matrix vesicles, which were never frozen nor lyophilized. Enzymatic activity, ion transports, ability to accumulate apatite in the lumen served to characterize matrix vesicles. It is correct that the addition of protectants, including trehalose, can contribute to preserve the enzymatic activity of membranous samples or membrane proteins after lyophilization or freeze process (As reported by JH Crowe, for instance). The problem with the protectant is that it is soluble, induce osmotic pressure, and leaks. At our knowledge, there are no papers reporting the possibility to preserve matrix vesicles after lyophilization. In addition, there are no papers using frozen matrix vesicles to investigate mineral property. The best conditions to preserve lyophilized or frozen matrix vesicles, need to be investigated. We added a short comment:

The conditions to preserve matrix vesicles and media vesicles need to be further optimized for possible therapeutical applications.

  • Each digestion method displays a different efficiency to release matrix vesicles and/or maintain their physical and biochemical properties. The authors should add a table describing these different methods with their advantages and disadvantages.

Author’s answer: We added a table describing different types of digestion methods (Table 1) and a short comment to support Table 1.

The collagenase-digestion method provides a relatively high amount of matrix vesicles displaying a TNAP specific activity of around 15 to 30 µmole min-1 mg-1 (Buchet et al. 2013). Alternatively, trypsin (Boyan et al. 1988), trypsin/collagenase (Kirsch et al. 1994), liberase/blendzyme-1 (Xiao et al. 2007), hyaluronidase (Boyan et al. 2022, Rosenthal et al. 2011), hyaluronidase/collagenase (Kirsch et al. 1994) can be used to release matrix vesicles from the extracellular matrix (Table 1). Collagenase digestion is among the most used digestion method to isolate matrix vesicles which can yield a high ratio of alkaline phosphatase activity in matrix vesicles compared to that in media vesicles or plasma membrane (Table 1). The collagenase and hyaluronidase digestion, which removed the surface collagens on the matrix vesicles induced a loss of calcium uptake (Kirsch et al. 1994) (Table 1) In contrast collagen and trypsin digestion which maintained a part of surface collagens on matrix vesicles preserved the calcium uptake (Kirsch et al. 1994) (Table 1). Trypsin digestion allowed to obtain matrix vesicles with six time higher alkaline phosphatase activity than that of plasma membrane (Skelton et al. 2023) (Table 1). Liberase and blendzyme1 which is gentle digestion, is the less efficient method to release matrix vesicles, as compared to other digestion methods, as reported by the lower ratio of alkaline phosphatase in matrix vesicles as compared to that in media vesicles (Xiao et al. 2007) (Table 1).

  • Figure 2A/B and Figure 6 have already been published previously by the authors (Bottini et al, Biochemica Biophysica Acta 2018).

Author’s answer: These figures are adapted from our papers and their reference are indicated in their figure-legend.

  • The content of extracellular vesicles appears to be dependent on the environment (inflammation, hypoxia). Do you know if such conditions can also influence matrix vesicles and modify their enzymatic activity? The authors should comment this point.

Author’s answer: We agree that there are many reports indicating that the content of extracellular vesicles are dependent of inflammation and hypoxia. In contrast, there are only very few reports on the influence of inflammation or hypoxia on the enzymatic activity of matrix vesicles. We added a short comment:

Not only osteogenic medium may affect the mineral property of the cells. Inflammatory cytokines can induce mineralization by influencing the enzymes regulating the pyrophosphate (inhibitor of mineralization) to phosphate ratio (Rosenthal 2016). Extracellular vesicles produced by IL-1β treated mesenchymal stem cells have altered ratio of ectonucleotide pyrophosphatase/phosphodiesterase 1 to alkaline phosphatase and less pyrophosphate in the vesicle fraction (Ferreira et al. 2013).

  • There are several examples that indicate changes in the properties of matrix vesicles, especially during the osteogenic differentiation. Cells (including chondrocytes and osteoblasts) which are not mineral competent release extracellular vesicles and matrix vesicles with little alkaline phosphatase activity and poor mineralization property, while once cells become fully mineralized they can release matrix vesicles with higher alkaline phosphatase activity and better inducer of mineral formation.

Author’s answer: Yes, that is correct. We added one paragraph:

Most of matrix vesicles with high TNAP activity are released from fully differentiated and mineral competent cells, while undifferentiated cells release extracellular vesicles with little TNAP activity (Bottini et al. 2018). Matrix vesicles are released from hypertrophic chondrocytes (Anderson 1969, Anderson 1995, Garimella et al. 2004, Kirsch and von der Mark 1992, Kirsch et al. 1997, Decalzi Cancedda et al. 1992). Hypertrophic chondrocyte differentiation is associated with high TNAP activity, and the synthesis and secretion of type X collagen followed by type II collagen by proliferating pre-hypertrophic chondrocytes (Kirsch et al. 2000, Anderson 1969, Anderson 1995, Garimella et al. 2004, Kirsch et al. 1992). Expression of type I collagen by hypertrophic chondrocytes might be associated with differentiation into osteoblast-like cells (Decalzi Cancedda et al. 1992, Gentili et al.1993, Roach et al. 1995). Matrix vesicles correspond to mineralizing cartilage, where the growing cartilage is replaced by bone, while articular cartilage matrix vesicles originate from normal cartilage, does not undergo matrix mineralization, except in pathological conditions such as osteoarthritis (Rosenthal et al. 2011). Articular matrix vesicles from normal cartilage  show low TNAP activity, however during osteoarthritis chondrocytes can become hypertrophic and fully mineralized (Rosenthal et al. 2011). They release articular matrix vesicles which can induce pathologic calcium crystal deposition in articular cartilage matrix (Rosenthal et al. 2016). The release of matrix vesicles by osteoblasts is stimulated in osteogenic medium containing ascorbic acid and beta-glycerophosphate (Dean et al. 1994). TNAP activity of Saos cells increased with the duration of the treatment with osteogenic factors  which coincided with the amount of released matrix vesicles (Thouverey et al. 2008).

6) Vascular calcification occurs when vascular smooth muscle cells or circulating cells differentiate to an osteogenic-like phenotype, synthesize an extracellular matrix and form apatite. However, vascular calcification often occurs with osteoporosis, a contradictory association called “calcification paradox”. In the study of Wang et al (Nat Com 2022), vesicles derived from aged bone matrix (AB-EVs) during bone resorption favor BMSC adipogenesis rather than osteogenesis and augment calcification of vascular smooth muscle cells. The authors should add this reference and comment matrix vesicle modifications with aging.

Author’s answer: Thank you for pointing attention to this reference. We added a comment: To explain why osteoporosis can contribute to vascular calcification -a calcification paradox-, it was reported that extracellular vesicles from aged bone matrix during bone resorption can favor the adipogenesis of bone marrow mesenchymal stem/stromal cells and increase vascular calcification (Wang et al. 2022). So far it is still unclear if such extracellular vesicles could originate from osteoclasts within the aged bone matrix. This opens the possibility that osteoclasts can release matrix-vesicle likes and media vesicles with distinct properties than those released from osteoblasts and from smooth muscle cells.

  • Extracellular vesicles, due to their favorable properties (biocompatibility, stability, low toxicity and exchange of molecular cargo) represent good candidates for tissue engineering and regenerative medicine. Matrix vesicles derived from mineralizing osteoblasts/mesenchymal stromal cells could be have considerable utility as an acellular approach to mineralized tissue engineering. The authors should comment this potential perspective.

Author’s answer: We added a comment:

Extracellular vesicles may represent good candidates for tissue engineering and regenerative medicines (Davies 2023, Ju et al 2022). In this respect matrix vesicles and media vesicles with their distinct properties derived from osteoblasts/mesenchymal stromal cells could have a considerable utility to mineralized tissue engineering. The conditions to preserve matrix vesicles and media vesicles need to be further optimized for possible therapeutical applications.

Reviewer 2 Report

Comments and Suggestions for Authors

Do media extracellular vesicles and extracellular vesicles bound to the extracellular matrix represent distinct types of vesicles?

A review by Saida Mebarek et al.

This is a comprehensive analysis of the differences between media extracellular vesicles and extracellular vesicles bound to the extracellular matrix.  This indeed is a timely review that underscores the significance of extracellular vesicles and their roles in mineralization competent cells.  The review has not only given a comprehensive historical perspective of the research in the field but more importantly the current research direction.  The only issue I had was with the numbering system of the sub-titles.  All of them are numbered 1.  For example, on page 1, line 33…1. Introduction.  However, on page 3, line 117 we also have…1. How to differentiate matrix…  On page 6, line 237… 1. Matrix vesicles and media vesicles from…. On page 9, line 350…. 1. Matrix vesicles and media vesicles from differentiated osteoblasts.  On page 11, line 441 ….1. Matrix vesicles and media vesicles from smooth muscle cells.  Lastly on page 12, line 477…. 1. Concluding remarks.  Please authors justify this numbering system.

Author Response

Reviewer 2: 
1)    This is a comprehensive analysis of the differences between media extracellular vesicles and extracellular vesicles bound to the extracellular matrix. This indeed is a timely review that underscores the significance of extracellular vesicles and their roles in mineralization competent cells. The review has not only given a comprehensive historical perspective of the research in the field but more importantly the current research direction. 

Author’s answer: Thank you for the comments. 

2)    The only issue I had was with the numbering system of the sub-titles. All of them are numbered 1. For example, on page 1, line 33…1. Introduction. However, on page 3, line 117 we also have…1. How to differentiate matrix… On page 6, line 237… 1. Matrix vesicles and media vesicles from…. On page 9, line 350…. 1. Matrix vesicles and media vesicles from differentiated osteoblasts. On page 11, line 441 ….1. Matrix vesicles and media vesicles from smooth muscle cells. Lastly on page 12, line 477…. 1. Concluding remarks. Please authors justify this numbering system.

Author’s answer: We corrected the numbering of chapters.